# LARGE LANGUAGE MODELS AS DECISION MAKERS FOR AUTONOMOUS DRIVING

## ABSTRACT

Existing learning-based autonomous driving (AD) systems face challenges in comprehending high-level information, generalizing to rare events, and providing interpretability. To address these problems, this work employs Large Language Models (LLMs) as a decision-making component for complex AD scenarios that require human commonsense understanding. We devise cognitive pathways to enable comprehensive reasoning with LLMs, and develop algorithms for translating LLM decisions into actionable driving commands. Through this approach, LLM decisions are seamlessly integrated with low-level controllers by guided parameter matrix adaptation. Extensive experiments demonstrate that our proposed method not only consistently surpasses baseline approaches in single-vehicle tasks, but also helps handle complex driving behaviors even multi-vehicle coordination, thanks to the commonsense reasoning capabilities of LLMs. This paper presents an initial step toward leveraging LLMs as effective decision-makers for intricate AD scenarios in terms of safety, efficiency, generalizability, and interoperability. We aspire for it to serve as inspiration for future research in this field. More demo videos and codes can be found at our anonymous webpage.

## 1 INTRODUCTION

Imagine you are behind the wheel, approaching an unsignalized intersection and planning to turn left, with an oncoming vehicle straight ahead. Human drivers intuitively know that according to traffic rules, they should slow down and yield, even if it is technically possible to speed through. However, existing advanced learning-based Autonomous Driving (AD) systems typically require complex rules or reward function designs to handle such scenarios effectively (Chen et al., 2023; Kiran et al., 2022). This reliance on predefined rule bases often limits their ability to generalize to various situations.

Another challenge facing existing learning-based AD systems is the long-tail problem (Buhet et al., 2019). Both limited datasets and sampling efficiency (Atakishiyev et al., 2023) can present challenges for existing learning-based AD systems when making decisions in rare real-world driving scenarios. Chauffeurnet (Bansal et al., 2018) demonstrated such limits where even 30 million state-action samples were insufficient to learn an optimal policy that mapped bird's-eye view images (states) to control (action).

Furthermore, the lack of interpretability (Gohel et al., 2021) is a pressing issue for existing learning-based AD systems. A mature AD system must possess interpretability to gain recognition within society and regulatory entities, allowing it to be subject to targeted optimization and iterative improvements. Nevertheless, existing learning-based AD systems inherently resemble black boxes, making it challenging to discern their decision-making processes or understand the rationale behind their actions (Atakishiyev et al., 2023). This lack of transparency can pose obstacles to the practical implementation of AD systems.

Considering the aforementioned challenges, a fundamental question arises: *Can we equip AD systems with the capability to think and drive like humans?* Our proposed solution involves employing a Large Language Model (LLM) to serve as the "brain" of the AD system. Recent introductions of models like ChatGPT (OpenAI, 2023), have positioned LLMs as early versions of Artificial General Intelligence (AGI) (Bubeck et al., 2023), owing to their remarkable emergent abilities (Wei et al., 2022) and innovative techniques such as Instruct Following and In-Context Learning (ICL) (Dong

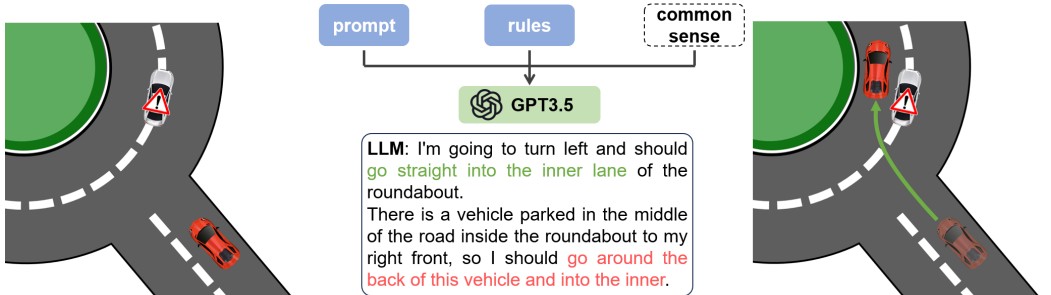

Figure 1: Faced with the rare situation where a breakdown vehicle is stopped in the middle of the road in a roundabout, LLM makes a decision that complies with traffic rules through common sense reasoning and understanding of high-level information.

et al., 2023). LLMs can think like humans, and reason about new scenarios by combining common sense, and the visible thinking process makes them strongly interpretable. These features make LLMs a powerful solution to the problems faced by AD systems described above.

In this paper, we leverage LLM to analyze and reason about various scenarios, enabling it to provide high-level decisions, and by tuning parameter matrix, we convert high-level decisions into mathematical representations to guide the bottom-level controller, Model Predictive Control (MPC). Fig. 1 illustrates the powerful reasoning capabilities of our system for rare and complex scenarios, demonstrating its superiority in understanding high-level information, commonsense reasoning, and interpretability. Through quantitative experiments, we showcase that our system significantly surpasses existing learning-based and optimization-based methods for single-vehicle decision-making tasks, with Overall Cost decreasing by **18.1%** and **16.4%**. Additionally, through qualitative experiments, we demonstrate the impressive capabilities of our system by effectively addressing intricate tasks, such as multi-vehicle joint control and driving behavior modulation guided by textual input.

The main contributions of this paper are as follows:

(1) We have devised a dedicated chain-of-thought framework for LLMs for driving scenarios, which divides the analysis and decision-making process into numerous sub-problems, enabling LLMs to comprehensively engage in logical reasoning and arrive at informed decisions.

(2) We have developed techniques for directing the bottom-level controller using high-level textual decisions provided by the LLM. This has enabled us to construct a comprehensive AD system that gives precise driving actions directly based on observational data.

(3) In a groundbreaking achievement, we have conducted quantitative experiments that conclusively showcase the substantial performance superiority of the AD system enhanced by LLM over existing methods. Additionally, we showcase our system's success in complex tasks, including coordinating multiple vehicles and regulating driving behavior with text-based input.

## 2 RELATED WORK

**Self-driving Autonomy.** Autonomous vehicles are governed by two primary paradigms: modular (Thrun et al., 2006; Daudelin et al., 2018) and end-to-end (Casas et al., 2021; Hu et al., 2023; Sadat et al., 2020; Tampuu et al., 2020). The modular approach involves a layered system of interconnected components responsible for different sub-tasks, including perception (Li et al., 2023; 2022b; Yin et al., 2021), planning (Kelly & Nagy, 2003; Zhang et al., 2022), and control (Peng et al., 2018; Johnson & Moradi, 2005). This design offers advantages such as modularity and versatility, but it comes with challenges related to tuning the pipeline and managing error propagation. In contrast, end-to-end autonomy directly translates sensor inputs into planner or controller commands. These methods are typically easier to develop but lack interpretability, making it challenging to diagnose errors, ensure safety, and incorporate traffic rules (Gohel et al., 2021). Nevertheless, recent progress in end-to-end learnable pipeline autonomy has shown promise by leveraging the strengths of both paradigms (Hu et al., 2023; Casas et al., 2021). Despite significant advancements in these self-driving paradigms, they often prove brittle when dealing with long-tail data or out-of-distribution scenarios encountered in real-world environments (Kong et al., 2023). This presents safety challenges in the context of autonomous driving.

**Advanced tasks with Large Language Models.** The success of Large Language Models (LLM) is undoubtedly exciting as it demonstrates the extent to which machines can learn human knowledge. Recent efforts in LLM have shown impressive performance in zero-shot prompting and complex reasoning (Bian et al., 2023; Nay, 2022; Chowdhery et al., 2022; Ouyang et al., 2022; Chung et al., 2022), embodied agent research (Wang et al., 2023a; Zhu et al., 2023; Vemprala et al., 2023; Driess et al., 2023; Yao et al., 2022) and addressing key transportation problems (Zheng et al., 2023), demonstrating its great ability to handle advanced tasks. In advanced tasks with LLMs, the translation of natural language input into actionable results is crucial. One prominent task is language-to-actions mapping, which has seen early approaches leveraging frameworks like temporal logic (Kress-Gazit et al., 2008) and motion primitive learning (Matuszek et al., 2013), evolving towards more recent end-to-end models for instruction-following in navigation (Ku et al., 2020) and manipulation tasks, employing latent embeddings of language commands (Jang et al., 2021; Mees et al., 2023; Lynch et al., 2022). Another critical dimension is language-to-code generation, extensively explored in contexts ranging from coding competitions (Li et al., 2022a) to instruction-following tasks (Liang et al., 2022). Moreover, the translation of natural language instructions into rewards has found applications in robotic domains, often requiring domain-specific reward models (Lin et al., 2022; Goyal et al., 2019; Nair et al., 2022). Kwon et al. (2023); Hu & Sadigh (2023) propose the use of LLMs for assigning reward values during Reinforcement Learning (RL) training. Additionally, incorporating iterative human feedback has been explored in correcting plans, with approaches employing semantic parsers (Broad et al., 2017) or trajectory optimization methods (Sharma et al., 2022). These various dimensions underscore the versatility and growing importance of LLMs in bridging the gap between natural language understanding and actionable outcomes in a wide range of applications.

Recent research has integrated LLMs into AD systems (Fu et al., 2023). However, in this work, the high-level safety judgments and decision-making processes rely on fixed rules rather than harnessing the full potential of LLMs' advanced reasoning capabilities, and the systems are currently limited to straightforward highway lane-changing scenarios. We aim to develop an AD system where LLMs play a central role in high-level decision-making. We extend their application to more intricate scenarios, such as navigating intersections and roundabouts, providing an initial step towards unlocking LLMs' effectiveness as decision-makers for complex AD scenarios.

## 3 METHOD

We develop an AD system with LLM as the core of high-level decision-making, as shown in Fig. 2(a). The LLM initiates a dialogue based on the provided prompt, continuously gathering information from the environment, engaging in reasoning, and rendering judgments. As illustrated in the center of Fig. 2(a), from left to right, the LLM proceeds sequentially: 1) identifies the vehicles requiring attention, 2) evaluates the situation, and 3) offers action guidance. Then the system transforms these three high-level textual decisions into mathematical representations, namely the observation matrix, weight matrix, and action bias. These elements serve as directives for the bottom-level controller, the MPC, instructing it on specific driving actions to be taken.

Taking the case of a left turn at an intersection as an example, in Fig. 2(b) we show how the three high-level textual decisions described above can be converted into the mathematical representations needed for MPC. The LLM selects 'vehicle_26', and we create a corresponding vector using the MPC's observation operator and zero out other elements in the observation matrix to focus solely on 'vehicle_26'. According to the waiting situation at the intersection that the LLM signals, we adjust the weight matrix to prioritize deceleration instructions over trajectory following, which prompts the MPC to decelerate promptly in alignment with the LLM's directive. We directly convert LLM's action guidance into action bias through predefined rules. Guided by the mathematical form of the above three aspects, the MPC completes the driving action of stopping and yielding.

### 3.1 BACKGROUND

The MPC solves a finite-time open-loop optimization problem online at each moment, based on the current measurement information obtained, and applies the first element of the resulting control sequence with the lowest cost to the controlled vehicle.

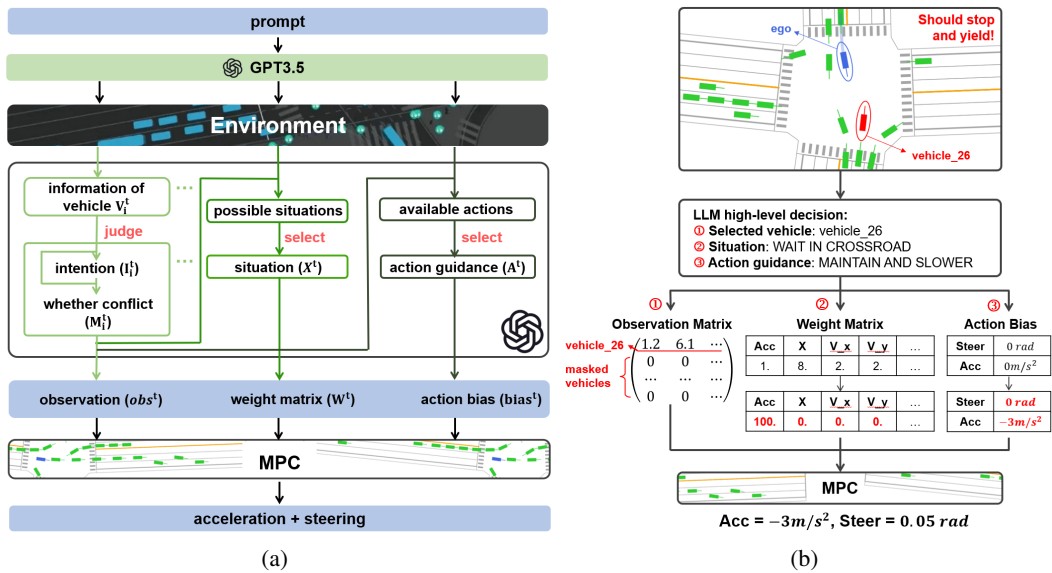

Figure 2: (a) Pipeline of our system with LLM as the core of high-level decision-making. (b) The LLM textual high-level decisions are converted into mathematical representations that guide the MPC to give specific driving actions. Take the case of a left turn at an intersection as an example.

In this work, we define the cost function of MPC in the context of Markov Decision Process (MDP), which is commonly used to formulate vehicle control problems: $(S, A, C, P, p_0)$, where $S$ is the state space, $A$ is the action space, $C : S \times A \mapsto \mathbb{R}$ is the cost function, $P : S \times A \mapsto S$ is the dynamics equation, and $p_0$ is the initial state distribution. Given a cost function $C$, the MPC finds a sequence of actions $\mathbf{a}_{1:H} = \mathbf{a}_1, \dots, \mathbf{a}_H$ that minimizes the expected accumulated cost $J(\mathbf{a}_{1:H}) = \sum_{t=1}^{H} C(\mathbf{s}_t, \mathbf{a}_t)$. The cost function takes the following form:

$$C(\mathbf{s}, \mathbf{a}) = \sum_{i=0}^{M} w_i \cdot \mathrm{n}_i\big(r_i(\mathbf{s}, \mathbf{a}, \psi_i)\big), \tag{1}$$

where $w \in \mathbb{R}+$ is a non-negative weight, $\mathrm{n}(\cdot) : \mathbb{R} \to \mathbb{R}+$ is a twice-differentiable norm that takes its minimum at 0, $r \in \mathbb{R}$ is a residual term that achieves optimality when $r = 0$, and $\psi_i$ is the parameters of the $i^{\text{th}}$ residual term. For example, if we want the vehicle to adopt the desired acceleration, we may design a residual term $r_{acc}(acc, \psi) = acc - \psi$, where the cost parameter $\psi$ denotes the desired acceleration, and use the $\ell_2$ norm to construct the final reward function: $C_{acc} = w|r_{acc}|_2$. Due to the complexity of driving scenarios, designing a set of weights and residual terms that are applicable to all driving scenarios is almost impossible (Askari et al., 2022). In this work, we use a generic and simple set of residual terms that include action biases to tune the control behavior, and design multiple sets of weight matrixes based on the certainty that the MPC should perform the action biases. We use the power of LLM to give action bias and select weight matrices for driving in complex scenarios.

## 3.2 CHAIN-OF-THOUGHT

We employ LangChain (Chase, 2023) as a framework to manage the LLM and establish a structured thought process for the LLM. This is achieved by defining a set of tools and specifying the sequence in which they should be utilized. To begin, we introduce these designated tools in a prompt at the outset of the conversation. Subsequently, during the course of the dialogue, the LLM actively invokes these tools to acquire pertinent information and guidance for its ongoing decision-making process. The LLM follows these guidelines to determine its next course of action until it successfully addresses the entire problem.

As an illustrative example, let's consider the three core tools depicted in Fig. 3. Each of these tools serves the dual purpose of providing the LLM with the relevant information and reasoning

Figure 3: Prompts for the three core tools we have defined. Each tool's prompt contains both information and guidelines to help LLM complete reasoning and judgment and begin the next step.

guidelines necessary to complete a specific reasoning step, while also directing the LLM on what actions it should take next.

Moreover, these tools enable us to revamp the way we deliver scenario information. Instead of overwhelming the LLM with all the scenario details at once, we provide only the relevant information needed for each decision step in the LLM's thought process, as depicted in Fig. 3. This approach represents a strategic shift that tackles a significant challenge faced by the LLM when dealing with complex and extensive data. It ensures that information is organized for simplicity and necessity, leading to a substantial improvement in the LLM's ability to reason and exercise judgment.

## 3.3 ATTENTION ALLOCATION

The ability to effectively distribute attention while driving reflects a human-like thought process. In this context, we task the LLM with systematically assessing information pertaining to surrounding vehicles, one at a time. Its objective is to discern the intentions of these vehicles and, ultimately, determine if they pose any conflicts with the movements of the ego vehicle. Specifically, At time $t$, for each element of surrounding vehicles $\mathcal{V}^t = \{V_1^t, V_2^t, \ldots\}$, we have:

$$I_i^t = \text{LLM}(S_i^t, S_i^{t-1}, \ldots, S_i^{t-10}, env^t), \tag{2}$$

$$M_i^t = \text{LLM}(S_i^t, I_i^t, env^t), \tag{3}$$

where $env^t$ is the scene road information, $S_i^t$ is the state of $V_i^t$ obtained from the environment, $I_i^t$ is the intention of $V_i^t$, and $M_i^t$ is 0 or 1, indicating whether the $V_i^t$ is considered by the LLM to be of concern. Subsequently, we create an observation matrix for the MPC exclusively based on the vehicles identified by the LLM:

$$obs_i^t = \text{MPC}_{\text{obs}}(S_i^t, I_i^t, env^t) * M_i^t, \tag{4}$$

where $obs_i^t$ is row $i$ of the MPC observation matrix, $\text{MPC}_{\text{obs}}$ is the operator for MPC to compute the observation matrix. This ensures that the MPC focuses solely on these selected vehicles.

## 3.4 SITUATION AWARENESS AND ACTION GUIDANCE

Situation awareness stands as a pivotal high-level decision-making process in driving, encompassing a profound understanding of the scenario at hand, along with common-sense reasoning. Within this framework, we tasked the LLM with the responsibility of selecting one specific situation from among several options, leveraging the information gleaned during the attention allocation process outlined in Section 3.3, as well as the results of the LLM's reasoned judgments. We define the feature $F_i^t = \{S_i^t, I_i^t, M_i^t\}$ to characterize the information associated with $V_i^t$, then we have:

$$X^t = \text{LLM}(F_1^t, \ldots, F_k^t, env^t), \tag{5}$$

where $k$ is the number of surrounding vehicles selected by LLM, and $X^t$ is the selected situation. The judgment of $X^t$ serves as a mechanism to tune the weight matrix of the MPC. For each predefined situation, we have established a corresponding weight matrix $W^t$. Subsequently, the LLM provided guidance concerning acceleration and steering based on its chosen situation:

$$A^t = \text{LLM}(F_1^t, \ldots, F_k^t, env^t, X^t), \tag{6}$$

Table 1: Evaluation on single-vehicle decision-making. SI refers to signalized intersection, USI refers to unsignalized intersection, and EOA refers to emergency obstacle avoidance. For all the metrics, the lower the better.

| Scenario | Method | Collision | Fail | Inefficiency | Time | Penalty | | Overall Cost |
|---|---|---|---|---|---|---|---|---|
| | | | | | | Acc | Dist | |
| SI | RL | 6 | 0 | 34.1 | **14.1** | 3.78 | 3.38 | 60.2 |
| | MPC | 2 | 4 | 25.5 | 17.7 | 3.14 | 3.05 | 56.3 |
| | Ours | **0** | **0** | **13.9** | 25.7 | **1.31** | **1.20** | **44.8** |
| USI | RL | 9 | **0** | 67.5 | **29.4** | 5.27 | 3.22 | 92.0 |
| | MPC | 2 | 4 | 74.0 | 30.7 | 4.30 | 2.55 | 87.2 |
| | Ours | **0** | 1 | **33.7** | 42.2 | **1.94** | **0.98** | **67.0** |
| Lane | RL | 0 | 0 | 2.27 | 6.8 | 0.15 | 0.09 | 8.71 |
| | MPC | 0 | 0 | 4.14 | 6.8 | 0.20 | 0.08 | 9.41 |
| | Ours | **0** | **0** | **1.13** | **6.7** | **0.08** | **0.03** | **7.63** |
| Roundabout | RL | 5 | 0 | 29.3 | **30.3** | 1.64 | 0.71 | 50.8 |
| | MPC | 1 | 3 | 29.3 | 30.4 | 1.61 | 0.68 | **50.6** |
| | Ours | **0** | **0** | **26.8** | 31.9 | **1.51** | **0.65** | 50.8 |
| EOA | RL | 11 | **0** | 32.3 | 16.9 | 2.99 | 1.96 | 51.3 |
| | MPC | 8 | 4 | 33.4 | 17.3 | 3.34 | 2.06 | 54.3 |
| | Ours | **3** | 2 | **28.8** | **16.7** | **2.60** | **1.79** | **47.3** |

where $A^t$ is action guidance. The $A^t$ influences the adjustment of the MPC's action bias, with costs decreasing as the vehicle's actions align more closely with the provided guidance:

$$r_{bias}(bias, \ \psi) = bias - \psi, \tag{7}$$

$$C_{bias} = w_{bias}|r_{bias}|_2, \tag{8}$$

where $bias$ is acceleration or steering. It's worth noting that our set of predefined situations, though limited in number, are abstract and broad enough to encompass a wide range of driving scenarios, as they do not represent specific scenarios, but rather the certainty that the MPC should perform the action biases. We substantiate the effectiveness of this approach through experimental validation in Section 4.1.

## 3.5 MULTI-VEHICLE JOINT CONTROL

Multi-vehicle joint control is an important solution for improving transportation efficiency and safety. However, both centralized and distributed approaches are often overly reliant on environmental prior and fail to exhibit good performance when the traffic model is unknown (Wang et al., 2023b). To address this problem, we propose a solution. Each vehicle is individually controlled by a distributed LLM, with one central LLM acting as the "brain" of the fleet for multi-vehicle communication and coordination. Each distributed LLM reports the situation it is into the central LLM and receives commands to control the ego vehicle; the central LLM judges and gives the coordination commands based on the environmental information and the reports from the distributed LLMs.

## 4 EXPERIMENTS

Our approach was applied to both single-vehicle decision-making and multi-vehicle joint control tasks. The scenario maps and traffic flows were generated using the IdSim (Liu et al., 2021).

In the context of single-vehicle decision-making, we conducted evaluations over three different approaches: Reinforcement Learning-Based Planning (RL) (Guan et al., 2023; Ren et al., 2022), Model Predictive Control (MPC) (Guan et al., 2023), and our system, MPC with LLM's High-Level Decision-Making Guidelines (LLM+MPC). Among them, RL was trained and validated in a wide range of complex traffic scenarios, and MPC was finely tuned and validated in real-vehicle experiments. Each of these approaches was tested across diverse scenarios, including signalized intersections, unsignalized intersections, driveways, emergency avoidance, and roundabouts. We selected 25 complex and challenging cases for each scenario type to comprehensively assess system performance. Evaluation metrics are detailed in Appendix A. Furthermore, We have also demonstrated the

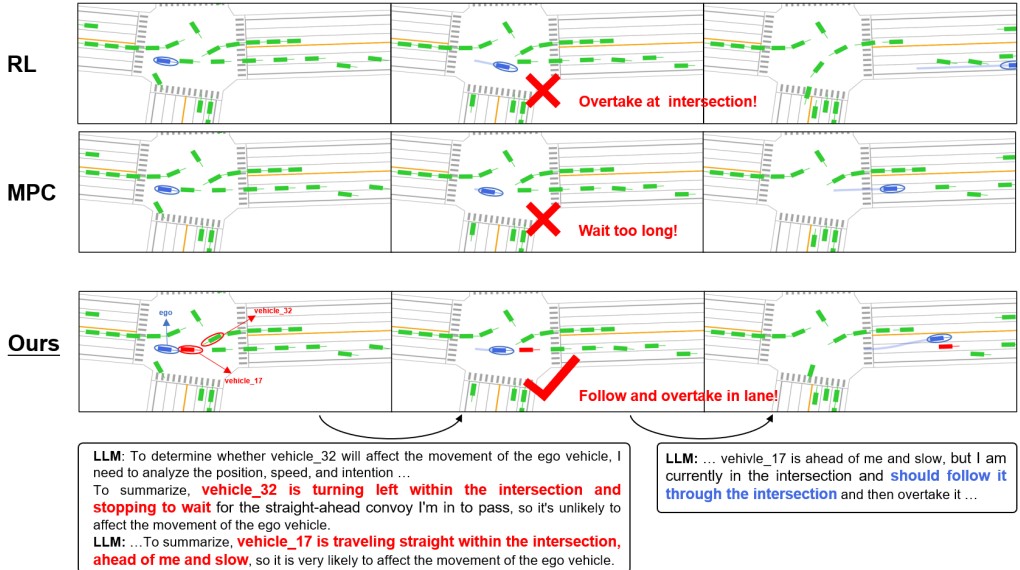

Figure 4: The ego vehicle is traveling straight through an unsignalized intersection. The red vehicle(s) in the last row is the one(s) selected by the LLM as needing attention. This example demonstrates LLM's understanding and reasoning about high-level information, proving the validity of the chain-of-thought we devised.

great potential of our system in driving behavior modulation guided by textual input, demonstrating its excellent understanding of high-level information.

In the realm of multi-vehicle joint control, our method's capabilities were put to the test in intricate gaming scenarios. Notably, we evaluated its performance in complex situations like narrow lane meetings, showcasing its adaptability and effectiveness in challenging environments.

### 4.1 SINGLE-VEHICLE DECISION-MAKING

The quantitative results of single-vehicle decision-making are shown in Table 1. It's important to highlight that our system achieves overall cost reductions across four scenario types, reflecting improved driving behavior. In non-emergency situations, we observe minimal fail occurrences and no collisions, underlining the safety of our approach. Additionally, in emergency scenarios, our method significantly lowers the accident rate, indicating its effectiveness in obstacle avoidance.

Specifically, in intersections, our focus is primarily on left-turn situations, where ego vehicle inherently possesses a lower right-of-way status, necessitating a deliberate choice to slow down and yield in accordance with established traffic rules. In left-turning within the intersections scenarios and roundabouts scenarios, although our approach may result in a slight increase in elapsed time, it yields substantial benefits in terms of enhanced traffic flow efficiency and reduced safety penalties. This outcome underscores our method's commitment to adopting safer and more reasonable driving behaviors that align with the principles of traffic regulations. In lanes, our approach excels in all metrics, indicating a more sensible approach to overtaking and lane changes. Finally, in emergency situations, our method demonstrates its effectiveness by reducing accident rates and enhancing overall performance.

**Attention Allocation.** Fig. 4 illustrates a scenario where the ego vehicle is proceeding straight through an unsignalized intersection. In this situation, the MPC incorporates all surrounding vehicles into its observation matrix for prediction and trajectory planning. However, it becomes evident from the outcomes that the MPC fails to accurately discern that "vehicle_32" has a lower right-of-way priority, erroneously decelerates and maneuvers to avoid it. In contrast, our approach, which employs LLM's reasoning, effectively comprehends the intention of "vehicle_32." As a result, it concentrates its attention solely on the foremost vehicle that could impact the ego vehicle's motion and appropriately follows it as it navigates the intersection. This example serves as a compelling demonstration of how the LLM in our approach adeptly comprehends the traffic scenario and discerns the

Table 2: Ablation experiments. OM refers to observation matrix; WM refers to weight matrix, and AB refers to action bias.

| Scenario | OM | WM&AB | Collision | Fail | Inefficiency | Time | Penalty Acc | Penalty Dist | Overall Cost |
|---|---|---|---|---|---|---|---|---|---|
| USI | × | × | 2 | 4 | 74.0 | 30.7 | 4.30 | 2.55 | 87.2 |
| | ✓ | × | 2 | 3 | 69.8 | **28.7** | 3.95 | 2.32 | 81.0 |
| | × | ✓ | **0** | 2 | 42.0 | 44.0 | 2.37 | 1.21 | 74.5 |
| | ✓ | ✓ | **0** | 1 | **33.7** | 42.2 | **1.94** | **0.98** | **67.0** |
| Roundabout | × | × | 1 | 3 | 29.3 | 30.4 | 1.61 | 0.68 | 50.6 |
| | ✓ | × | 1 | 3 | 29.4 | **30.3** | 1.61 | 0.69 | 50.6 |
| | × | ✓ | 0 | 0 | 27.8 | 30.9 | 1.53 | 0.67 | **50.2** |
| | ✓ | ✓ | **0** | **0** | **26.8** | 31.9 | 1.51 | **0.65** | 50.8 |

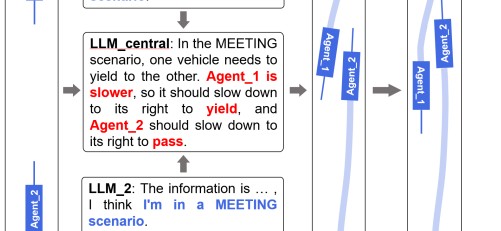

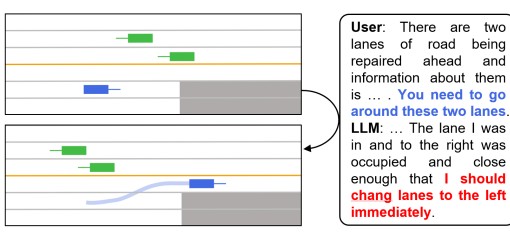

(a) Multi-vehicle joint control  (b) Textual guidance for complex scenarios

Figure 5: (a) The convoy is in a meeting situation. This example demonstrates that the high-level decision-making of the central LLM and the fine-grained control of the distributed LLMs collaborate to accomplish high-quality multi-vehicle cooperative control. (b) LLM changes lanes to avoid road construction under textual guidance.

intentions of other vehicles. This enables it to make more informed decisions about allocating attention and, consequently, facilitates more efficient and rational driving behavior.

**Situation Awareness and Action Guidance.** In the scenario depicted in Fig. 4, a crucial factor to consider is that the ego vehicle, despite being hindered by the slower-moving vehicle ahead, is obligated by traffic regulations not to perform overtaking maneuvers within the intersection. However, the RL opts to overtake the leading vehicle from the right side immediately within the intersection. This decision stems from the inherent limitations of learning-based approaches, which often struggle to grasp high-level information such as traffic regulations. Their primary focus tends to be on achieving a certain level of efficiency and safety in driving. In contrast, our approach, as indicated in the dialogue box on the right, showcases the LLM's capacity to accurately comprehend the ego vehicle's situation. It makes a reasoned choice to follow the vehicle ahead through the intersection before considering an overtaking maneuver, which aligns with traffic regulations. This example underscores the LLM's proficiency in reasoning about complex traffic scenarios and assessing its own circumstances. As a result, it produces decisions that closely mimic human thinking, guided by higher-level information like traffic regulations and common sense.

**Ablations.** We conducted ablation experiments in two typical driving scenarios, unsignalized intersections, and roundabouts, and the outcomes are presented in Table 2. When solely employing the LLM's Attention Allocation capability, nearly all metrics exhibited improvement compared to the baseline MPC. This improvement stemmed from the system's ability to disregard irrelevant surrounding vehicles, leading to more effective decision-making. On the other hand, when utilizing only the LLM Situation Awareness and Action Guidance features, all metrics, except for time, demonstrated substantial enhancement compared to the baseline MPC. This highlights the pivotal role played by our system's comprehension of high-level information in decision-making.

## 4.2 MULTI-VEHICLE JOINT CONTROL

We have assessed the effectiveness of our approach through a specific scenario involving a narrow road encounter, as depicted in Fig. 5(a). In this scenario, two distributed LLMs concurrently report

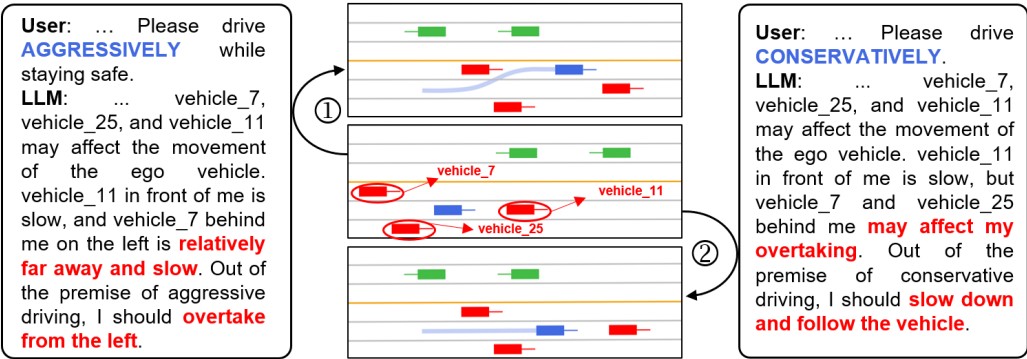

Figure 6: The ego vehicle is traveling in lane, the front vehicle is slow, and the ego vehicle overtaking would be low risk. This example demonstrates the ease and effectiveness of LLM in driving style adjustment.

that they are in a meeting situation. The central LLM, informed by the scenario details and the self-assessment of each distributed LLM, issues instructions. One vehicle is instructed to decelerate and wait, while the other is advised to slow down and proceed, thus facilitating communication and coordination within the convoy. Subsequently, each distributed LLM adjusts the control of its respective vehicle in accordance with this convoy-level decision. This example vividly illustrates how our system combines the strengths of both centralized and distributed methodologies. The central LLM acts as the "brain" for convoy communication and coordination, while distributed LLMs can intelligently manage their respective vehicles based on decisions made at the convoy level.

### 4.3 TEXT-MODULATED DRIVING BEHAVIOR

**Driving style adjustment.** In real-world driving scenarios, users often desire the ability to effortlessly customize the driving behavior of AD systems to align with their preferences for efficiency and comfort. However, for learning-based or optimization-based AD systems, achieving this level of intuitive and reliable customization requires complex rule or reward function designs (Chang et al., 2023). In contrast, our approach simplifies the process by merely providing textual descriptions to the LLM through a dedicated interface. Fig. 6 exemplifies this feature. When there is low risk of overtaking, LLM instructed to drive aggressively will make reasonable overtaking decisions, while those directed to drive conservatively will opt to slow down and follow the vehicle in front of it. This example effectively illustrates how our approach excels at comprehending the user's abstract and non-intuitive requirements, easily delivering the expected driving behavior.

**Textual guidance for complex scenarios.** Certain complex transportation scenarios, such as road construction and other uncommon situations, pose significant challenges for many existing AD systems (Chen et al., 2023). However, these scenarios are typically straightforward for humans to identify and understand. To address this issue, our approach enables users or utilizes high-precision maps to provide textual instructions that guide the AD system's decision-making process. As depicted in Fig. 5(b), we conducted an experiment involving a road construction scenario. Upon receiving textual guidelines, our approach successfully recognized the situation and gave appropriate driving behavior.

### 5 CONCLUSION

This paper demonstrates that LLMs can effectively serve as the core high-level decision-making component of AD systems. Our approach combining LLMs and MPC substantially outperforms existing methods on key metrics and handles complex real-world driving scenarios. The reasoning skills and interpretability of LLMs help overcome limitations of current learning-based AD systems regarding adaptability and transparency. This paper makes a compelling case for LLMs as a transformative solution to enable human-like performance in diverse driving scenarios. Our approach provides an initial step for developing safe, efficient, generalizable, and interpretable LLM-based AD systems. We aspire for it to serve as inspiration for future research in this domain.

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

## A METRICS

We use the metrics below to measure the safety and efficiency of driving behavior.

**Number of failure/collision cases:** We keep a tally of failure cases. A case is marked as a failure if the ego vehicle cannot reach its target area within a designated 200-second time frame. For clarity, when the ego vehicle travels within a lane, the target area is defined as the end of that lane. When it navigates through an intersection or roundabout, the target area is set as the beginning of the target lane. Collision cases are tracked separately.

**Inefficiency:** To gauge the efficiency of traffic flow, we calculate the average difference between the ego vehicle's maximum and current speeds:

$$\xi = \frac{1}{N} \sum_{i=1}^{N} (v_i^{max} - v_i). \tag{9}$$

Our assessment focuses solely on the lead vehicle within each convoy, as it is the one directly influenced by the ego vehicle's driving behavior. Vehicles affected by red lights are excluded from this calculation.

**Time:** We directly record the time the ego vehicle takes to reach its target area. This metric serves as an indicator of the ego vehicle's driving efficiency.

**Penalty:** The penalty metric is employed to assess the safety of the ego vehicle's driving behavior. It's computed based on the distance between the target vehicle and the ego vehicle, as well as the deceleration of the target vehicle. Smaller distances and higher decelerations indicate more unsafe driving behavior, thus contributing to a higher penalty:

$$P_{dec} = w_{dec} \sum_{i=1}^{N} f_{dec}(dec_i - dec_0), \tag{10}$$

$$P_{dist} = w_{dist} \sum_{i=1}^{N} f_{dist}(dist_0 - dist_i, \ dec_i - dec_0), \tag{11}$$

$$f_{dec}(x) = \begin{cases} 0 & \text{if } x \leq 0, \\ x & \text{if } x > 0, \end{cases} \quad f_{dist}(x, y) = \begin{cases} 0 & \text{if } y \leq 0, \\ x & \text{if } y > 0, \end{cases} \tag{12}$$

where $dec$ refers to decceleration, $dist$ refers to distance, $w_{dec} = 100$, $w_{dist} = 1$, $dec_0 = 1m/s^2$, and $dist_0 = 50m$. This penalty is calculated for each vehicle, similar to the Efficiency metric, and then aggregated to derive the final penalty score.

**Cost:** To provide an overall evaluation, we apply weighted values to the above metrics so that the values of each item are close to each other:

$$Cost = 30 * \xi + t + 50 * P_{dec} + 5 * P_{dist}. \tag{13}$$

This comprehensive cost assessment enables us to holistically evaluate the performance of our autonomous vehicle system.

# B  DECISION-MAKING PROCESS

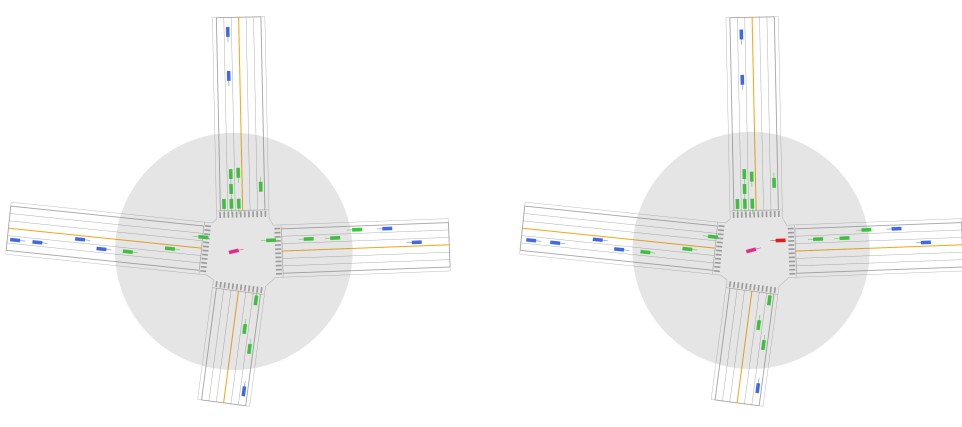

(a) Scenario                    (b) Driving behavior of LLM+MPC

Figure 7: Pink one is the ego vehicle, green ones are the vehicles within the perception range of the ego vehicle, blue ones are the vehicles outside the perception range of the ego vehicle, and red one is the vehicle chosen by the LLM.

The complete decision-making process of LLM is shown in the text box below. Faced with the scenario in Fig. 7(a), LLM+MPC gives the driving actions shown in Fig. 7(b).

**Prompt**:
You, the 'ego' car, are now driving a car.
The situation and action you chose LAST time step were 'CROSS_NORMAL' and 'SLIGHTLY_LEFT_SLOWER'.

The information about ego vehicle is as follows:
The speed of the ego vehicle is 8.0m/s. There is no traffic light in front of the ego vehicle.
The ego vehicle is making a left turn in the intersection.
The ego vehicle is coming from lane_-17_4. The start of lane_-17_4 is on ego vehicle's counterclockwise rotation of 168 degrees, 16.4m away, its direction is ego vehicle rotated clockwise by 18 degrees. lane_-17_4 is for left turns ONLY.
The ego vehicle is heading onto road_3. The start of road_3 is on ego vehicle's counterclockwise rotation of 43 degrees, 25.1m away, its direction is ego vehicle rotated counterclockwise by 78 degrees.
If ignoring any surrounding traffic participants but respecting traffic lights, the ego vehicle will pass these given waypoints:
waypoint_0: it is on ego vehicle's counterclockwise rotation of 23 degrees, 9.6m away.
waypoint_1: it is on ego vehicle's counterclockwise rotation of 44 degrees, 18.3m away.

The intersection has 8 roads.
The information of roads is as follows (where the direction of the road/lane is meant to be the direction in which the vehicles on the road are permitted to travel):
Start of road_-17 is on ego vehicle's clockwise rotation of 168 degrees, 18.8m away, its direction is ego vehicle rotated clockwise by 18 degrees. road_-17 contains three lanes: lane_-17_2, lane_-17_3, lane_-17_4.
Start of road_-3 is on ego vehicle's counterclockwise rotation of 90 degrees, 21.1m away, its direction is ego vehicle rotated clockwise by 101 degrees. road_-3 contains three lanes: lane_-3_2, lane_-3_3, lane_-3_4.
Start of road_-7 is on ego vehicle's clockwise rotation of 74 degrees, 24.3m away, its

direction is ego vehicle rotated counterclockwise by 69 degrees. road_-7 contains three lanes: lane_-7_2, lane_-7_3, lane_-7_4.

Start of road_-9 is on ego vehicle's counterclockwise rotation of 8 degrees, 25.9m away, its direction is ego vehicle rotated counterclockwise by 169 degrees. road_-9 contains three lanes: lane_-9_2, lane_-9_3, lane_-9_4.

Start of road_17 is on ego vehicle's counterclockwise rotation of 131 degrees, 18.8m away, its direction is ego vehicle rotated counterclockwise by 161 degrees. road_17 contains three lanes: lane_17_2, lane_17_3, lane_17_4.

Start of road_3 is on ego vehicle's counterclockwise rotation of 43 degrees, 25.1m away, its direction is ego vehicle rotated counterclockwise by 78 degrees. road_3 contains three lanes: lane_3_2, lane_3_3, lane_3_4.

Start of road_7 is on ego vehicle's clockwise rotation of 122 degrees, 20.1m away, its direction is ego vehicle rotated clockwise by 110 degrees. road_7 contains three lanes: lane_7_2, lane_7_3, lane_7_4.

Start of road_9 is on ego vehicle's clockwise rotation of 33 degrees, 26.4m away, its direction is ego vehicle rotated clockwise by 10 degrees. road_9 contains three lanes: lane_9_2, lane_9_3, lane_9_4.

Please select traffic participants who affect the movement of ego vehicle and record them, then determine the situation you're in, and finally give action guidance for the ego vehicle.
Follow these steps:
1. Get the information of ONE specific traffic participant.
2. After get the information of that traffic participant, determine whether or not it will affect the movement of the ego vehicle and give reasons.
3. Use tool 'Record Judgments' to record the judgement.
4. Please COMPLETE the three steps for ONE traffic participant BEFORE focusing on the next traffic participant. Repeat the above three steps until you have checked all traffic participants.
5. After you have checked all traffic participants, analyze the possible situations you might be in and select ONE from them, and use 'Record Situation' to record it.
6. Analyze the possible actions and select ONE from them, and use 'Record Action' to record it.
7. Output the final answer.
Before giving your final answer, MAKE SURE you have completed all of the above steps.

The final answer should contain the following four parts:
1. Select traffic participants who affect the movement of ego vehicle.
2. The situation you're in. Make sure you've used tool 'Get Possible Situations' to get the possible situations, from which the situation you answer must be selected.
3. Action to be taken by the ego vehicle. Make sure you've used tool 'Get Available Actions' to get the available actions, from which the action you answer must be selected.
4. Your explanation about your decision on the above three things.

Here is an example of using a tool:
...
Thought: To determine which traffic participants affect the movement of the ego vehicle, I need to get the IDs of all traffic participants and analyze their states.

Action:
```
{
"action": "Get IDs of all Traffic Participants",
"action_input": "ego"
}
```
...

After this, you need to wait for the answer before taking the next step.

Remember the conversations you've had and DONOT repeat them.
Let's think step by step. Remember to use the proper tools mentioned in the tool list ONCE a time. FOLLOW THE TOOLS' GUIDELINES FOR YOUR NEXT STEP!
DONOT give the final answer now untill you are asked to! The final answer should be in the following format:

...
Final Answer:
"participants":{"vehicles who affect the movement of ego vehicle, a list of their ids"},
"situation":{the situation you're in, ONE of the possible situations that tools give you},
"action":{"action to be taken by the ego vehicle, ONE of the available actions"}

...
END EACH OF YOUR ANSWERS WITH THE USE OF ONE TOOL, unless you are asked to give the final answer.
When you are asked to use a tool, you MUST use it at the end of the next answer!
First use tool 'Get IDs of all Traffic Participants'!

**LLM**:
To determine which traffic participants affect the movement of the ego vehicle, I need to get the IDs of all traffic participants and analyze their states.
Action:
```
{
"action": "Get IDs of all Traffic Participants",
"action_input": "ego"
}
```

**Prompt**:
There are 1 traffic participants:
'vehicle_15'.
DONOT use this tool again!
Now use tool 'Get Information about Traffic Participant' with input 'vehicle_15' to get its information.

**LLM**:
To determine whether or not the traffic participant with ID 'vehicle_15' will affect the movement of the ego vehicle, I need to get its information and analyze its state.
Action:
```

"action": "Get Information about Traffic Participant",
"action_input": "vehicle_15"

```

**Prompt**:
The information about vehicle_15 is: vehicle_15 is on ego vehicle's counterclockwise rotation of 3 degrees, 19.7m away.
The absolute velocity of vehicle_15 is 4.0m/s and the direction is ego vehicle rotated counterclockwise by 169 degrees. The direction is roughly towards the ego vehicle. The

velocity of vehicle_15 relative to ego vehicle is 11.9m/s and the direction is ego vehicle rotated counterclockwise by 176 degrees.

vehicle_15 came from lane_-9_3. Start of lane_-9_3 is on ego vehicle's counterclockwise rotation of 0 degrees, 24.9m away, its direction is ego vehicle rotated counterclockwise by 169 degrees. lane_-9_3 is straight ahead ONLY.

In last 10 time steps (1 time step is 0.1 second), the position and absolute velocity of vehicle_15 is as follows:

t=-1, vehicle_15 is on ego vehicle's counterclockwise rotation of 2.7 degrees, 20.1m away. Its absolute velocity is 4.0m/s and the direction is ego vehicle rotated counterclockwise by 169 degrees.

t=-2, vehicle_15 is on ego vehicle's counterclockwise rotation of 2.5 degrees, 20.5m away. Its absolute velocity is 4.0m/s and the direction is ego vehicle rotated counterclockwise by 169 degrees.

t=-3, vehicle_15 is on ego vehicle's counterclockwise rotation of 2.2 degrees, 20.9m away. Its absolute velocity is 4.0m/s and the direction is ego vehicle rotated counterclockwise by 169 degrees.

t=-4, vehicle_15 is on ego vehicle's counterclockwise rotation of 2.0 degrees, 21.3m away. Its absolute velocity is 4.0m/s and the direction is ego vehicle rotated counterclockwise by 169 degrees.

t=-5, vehicle_15 is on ego vehicle's counterclockwise rotation of 1.7 degrees, 21.6m away. Its absolute velocity is 4.0m/s and the direction is ego vehicle rotated counterclockwise by 169 degrees.

t=-6, vehicle_15 is on ego vehicle's counterclockwise rotation of 1.5 degrees, 22.0m away. Its absolute velocity is 4.0m/s and the direction is ego vehicle rotated counterclockwise by 169 degrees.

t=-7, vehicle_15 is on ego vehicle's counterclockwise rotation of 1.3 degrees, 22.4m away. Its absolute velocity is 4.0m/s and the direction is ego vehicle rotated counterclockwise by 169 degrees.

t=-8, vehicle_15 is on ego vehicle's counterclockwise rotation of 1.1 degrees, 22.8m away. Its absolute velocity is 4.0m/s and the direction is ego vehicle rotated counterclockwise by 169 degrees.

t=-9, vehicle_15 is on ego vehicle's counterclockwise rotation of 0.8 degrees, 23.2m away. Its absolute velocity is 4.0m/s and the direction is ego vehicle rotated counterclockwise by 169 degrees.

t=-10, vehicle_15 is on ego vehicle's counterclockwise rotation of 0.7 degrees, 23.6m away. Its absolute velocity is 4.0m/s and the direction is ego vehicle rotated counterclockwise by 169 degrees.

The speed of the ego vehicle is 8.0m/s, it is making a left turn in the cross road.

The ego vehicle came from lane_-17_4. Start of lane_-17_4 is on ego vehicle's counterclockwise rotation of 168 degrees, 16.4m away, its direction is ego vehicle rotated clockwise by 18 degrees. lane_-17_4 is for left turns ONLY.

The ego vehicle is heading onto road_3. Start of road_3 is on ego vehicle's counterclockwise rotation of 43 degrees, 25.1m away, its direction is ego vehicle rotated counterclockwise by 78 degrees.

The ego vehicle's reference trajectory is given by the form of waypoints, i.e., if ignoring any surrounding traffic participants but respecting traffic lights, the ego vehicle will pass these given waypoints:

waypoint_0: it is on ego vehicle's counterclockwise rotation of 23 degrees, 9.6m away.

waypoint_1: it is on ego vehicle's counterclockwise rotation of 44 degrees, 18.3m away.

First you need to determine the intention of vehicle_15 and give reasons.

You need to follow these steps to give the above judgment:

1. Consider the state of vehicle_15 in lase 10 time steps and now to determine the intention of vehicle_15. Give reasons.

2. Take the location and orientation of the road network into account. Give the reasons.

3. Summarize the intention of vehicle_15. Do not give reasons here.

Then you need to determine whether or not vehicle_15 will affect the movement of the ego vehicle and give reasons.

You need to follow these steps to give the above judgment:

1. Determine whether the position of vehicle_15 is likely to affect the motion of the ego vehicle along the reference trajectory. Give the reasons.

2. Consider both the velocity and position of vehicle_15 and the velocity of ego vehicle to determine whether it affects the motion of the ego vehicle along the reference trajectory. Give the reasons.

3. Take intention of vehicle_15 and ego vehicle into account. Give the reasons.

After the above judgment, use tool 'Record Judgments' to record the judgment with input 'vehicle_15,{str},{num}', where str is a string no more than five words, and num is 0 or 1, with 0 representing no and 1 representing yes. str represents the intention of vehicle_15, and num represents whether or not vehicle_15 will affect the movement of the ego vehicle.

DONOT use tool 'Get Information about Traffic Participant' UNTILL you have made above judgments for vehicle_15 and used tool 'Record Judgments' to record the judgment!

**LLM**:

To determine the intention of vehicle_15, I need to analyze its state and road network.

1. For the last 10 time steps and the present, the direction of vehicle_15's speed is ego vehicle rotated counterclockwise by 169 degrees, which has not changed, so the trajectory is roughly along a straight line. Therefore vehicle_15 is traveling in a straight line.

2. vehicle_15 came from lane_-9_3, which is straight ahead only. The direction of lane_-9_3 is ego vehicle rotated counterclockwise by 169 degrees, and direction of the velocity of vehicle_15 is ego vehicle rotated counterclockwise by 169 degrees. vehicle_15 is traveling roughly in the direction of lane_-9_3. Also, start of lane_-9_3 is on ego vehicle's counterclockwise rotation of 0 degrees, 24.9m away, and ego vehicle came from lane_-17_4 whose start is on ego vehicle's counterclockwise rotation of 168 degrees, 16.4m away and direction is ego vehicle rotated counterclockwise by 78 degrees. lane_-9_3 may be the opposite lane of lane_-17_4. Therefore vehicle_15 is a straight ahead vehicle from the opposite lane of the ego vehicle.

3. To summarize, vehicle_15 is from the opposite lane of the ego vehicle, and its intention is to go straight.

To determine whether or not vehicle_15 will affect the movement of the ego vehicle, I need to analyze its position, velocity relative to the ego vehicle, and the intention of vehicle_15 and ego vehicle.

1. The position of vehicle_15 is on the left front of the ego vehicle. Since the ego vehicle is making a left turn, the position of vehicle_15 is likely to affect the motion of the ego vehicle along the reference trajectory. This is because vehicle_15 is in a position where it could potentially intersect with the path of the ego vehicle.

2. Considering the velocity and position of vehicle_15, as well as the velocity of the ego vehicle, it is possible that vehicle_15 could affect the motion of the ego vehicle. The absolute velocity of vehicle_15 is 4.0m/s, and its direction is roughly towards the ego vehicle. The velocity of vehicle_15 relative to the ego vehicle is 11.9m/s, and its direction is roughly towards the ego vehicle as well. This indicates that vehicle_15 is moving towards the ego vehicle at a relatively high speed. Given the relative positions and velocities, there is a possibility of a collision or interference between the two vehicles.

3. vehicle_15 is from the opposite lane of the ego vehicle, and its intention is to go straight. So it's quite possible that it will affect the movement of the ego vehicle.

Now I will use the tool 'Record Judgments' to record the judgment for vehicle_15.

Action:
```
{

"action": "Record Judgments",
"action_input": "vehicle_15,go straight from opposite lane,1"
}
"'

**Prompt**:
You have judged all traffic participants one by one to determine their intentions and whether they will affect the movement of the ego vehicle. Now use tool 'Get Possible Situations'.

**LLM**:
To determine the possible situations I might be in, I need to use the tool 'Get Possible Situations'.
Action:
"'
{
"action": "Get Possible Situations",
"action_input": "ego"
}
"'

**Prompt**:
There are 1 traffic participants you need to consider: 'vehicle_15'.
The information about vehicle_15 is:
vehicle_15 is on ego vehicle's counterclockwise rotation of 3 degrees, 19.7m away.
The absolute velocity of vehicle_15 is 4.0m/s and the direction is ego vehicle rotated counterclockwise by 169 degrees. The direction is roughly towards the ego vehicle.
The velocity of vehicle_15 relative to ego vehicle is 11.9m/s and the direction is ego vehicle rotated counterclockwise by 176 degrees.
vehicle_15 is coming from lane_-9_3. Start of lane_-9_3 is on ego vehicle's counterclockwise rotation of 0 degrees, 24.9m away, its direction is ego vehicle rotated counterclockwise by 169 degrees. lane_-9_3 is straight ahead ONLY.
The intention of vehicle_15 is go straight from opposite lane.

The information about ego vehicle is:
The speed of the ego vehicle is 8.0m/s. There is no traffic light in front of the ego vehicle.
The ego vehicle is making a left turn in the crossroad.
The ego vehicle is coming from lane_-17_4. Start of lane_-17_4 is on ego vehicle's counterclockwise rotation of 168 degrees, 16.4m away, its direction is ego vehicle rotated clockwise by 18 degrees. lane_-17_4 is for left turns ONLY.
The ego vehicle is heading onto road_3. Start of road_3 is on ego vehicle's counterclockwise rotation of 43 degrees, 25.1m away, its direction is ego vehicle rotated counterclockwise by 78 degrees.
If ignoring any surrounding traffic participants but respecting traffic lights, the ego vehicle will pass these given waypoints:
waypoint_0: it is on ego vehicle's counterclockwise rotation of 23 degrees, 9.6m away.
waypoint_1: it is on ego vehicle's counterclockwise rotation of 44 degrees, 18.3m away.

You can ONLY choose ONE of the following situations:
CROSS_NORMAL–this is a normal situation, you only need to follow the reference trajectory and DONOT need to slow down;
CROSS_WAIT–in this case it would be unsafe for you to go through the crossroad without slowing down, therefore you need to slow down and wait;
CROSS_FAST–in this case you need to speed up and rush through the crossroad considering safety and efficiency;

'CROSS_NORMAL' should be prioritized when you believe it is safe to go through the crossroad without slowing down.
'CROSS_WAIT' is for safety and traffic regulations, prioritize it when you believe it is NOT safe to go through the crossroad without slowing down.
'CROSS_FAST' is for efficiency, please make sure it's safe!
Note that left-turning vehicles in an intersection must yield to oncoming straight traffic!

DONOT use this tool again and DONOT give the final answer now untill you are asked to!
Now determine which of the above situations you are in and then use tool 'Record Situation' to record it!

**LLM**:
To determine the possible situations I might be in, I need to consider the information about the ego vehicle and the traffic participant vehicle_15. Based on the information provided, I can analyze the situation and make a decision.

Considering the position, velocity and intention of vehicle_15, as well as the speed and trajectory of the ego vehicle, I believe the possible situation I am in is CROSS_WAIT. This is because the ego vehicle is turning left in an intersection and vehicle_15 is a straight ahead vehicle from the opposite lane of the ego vehicle, so the ego vehicle must yield. Also, there is a possibility of a collision or interference between vehicle_15 and the ego vehicle. Therefore, it would be unsafe for the ego vehicle to go through the crossroad without slowing down.

Now I will use the tool 'Record Situation' to record the situation as CROSS_WAIT.
Action:
```
{
"action": "Record Situation",
"action_input": "CROSS_WAIT"
}
```

**Prompt**:
Now use tool 'Get Available Actions'!

**LLM**:
To determine the available actions for the ego vehicle in the CROSS_WAIT situation, I need to use the tool 'Get Available Actions'.
Action:
```

"action": "Get Available Actions",
"action_input": "ego"

```

**Prompt**:
There are 1 traffic participants you need to consider: 'vehicle_15'.

The situation you are in is 'CROSS_WAIT'.
You can ONLY use one of the following actions:

SLIGHTLY_LEFT_FASTER–steer left between 3 and 10 degrees, and accelerate the vehicle;
SHARPLY_LEFT_FASTER–steer left 10 degrees or more, and accelerate the vehicle;
SLIGHTLY_LEFT_MAINTAIN–steer left between 3 and 10 degrees with current speed;
SHARPLY_LEFT_MAINTAIN–steer left 10 degrees or more with current speed;
SLIGHTLY_LEFT_SLOWER–steer left between 3 and 10 degrees, and decelerate the vehicle;
SHARPLY_LEFT_SLOWER–steer left 10 degrees or more, and decelerate the vehicle;
SLIGHTLY_RIGHT_FASTER–steer right between 3 and 10 degrees, and accelerate the vehicle;
SHARPLY_RIGHT_FASTER–steer right 10 degrees or more, and accelerate the vehicle;
SLIGHTLY_RIGHT_MAINTAIN–steer right between 3 and 10 degrees with current speed;
SHARPLY_RIGHT_MAINTAIN–steer right 10 degrees or more with current speed;
SLIGHTLY_RIGHT_SLOWER–steer right between 3 and 10 degrees, and decelerate the vehicle;
SHARPLY_RIGHT_SLOWER–steer right 10 degrees or more, and decelerate the vehicle;
IDLE_FASTER–steering within 3 degrees to the left or right, and accelerate the vehicle;
IDLE_MAINTAIN–steering within 3 degrees to the left or right with current speed;
IDLE_SLOWER–steering within 3 degrees to the left or right, and decelerate the vehicle;

Prioritize actions that are more consistent with the reference trajectory (i.e., waypoints of the ego vehicle). Consider other actions ONLY when you believe they will result in a collision or are unsafe.
Prioritize acceleration and maintaining speed. Consider other deceleration ONLY when you believe they will result in a collision or are unsafe.
DONOT use this tool again and DONOT give the final answer now untill you are asked to!
Now select ONE action and then use tool 'Record Action' to record it!

**LLM**:
To determine the available actions for the ego vehicle in the CROSS_WAIT situation, I need to consider the position and velocity of the traffic participant vehicle_15, as well as the speed and trajectory of the ego vehicle.

I would prioritize actions that are more consistent with the reference trajectory of the ego vehicle, such as SLIGHTLY_LEFT_MAINTAIN or SLIGHTLY_RIGHT_SLOWER. Considering the situation I am in is CROSS_WAIT, I should choose SLIGHTLY_RIGHT_SLOWER. This actions allow the ego vehicle to slow down and steer slightly to the left, which can help avoid a potential collision with vehicle_15.

Now I will use the tool 'Record Action' to record the action as SLIGHTLY_LEFT_SLOWER.
Action:
```
{
"action": "Record Action",
"action_input": "SLIGHTLY_LEFT_SLOWER"
}
```

**Prompt**:
Now summerize the final answer!

**LLM**:
Final Answer:

```
{
"participants": "vehicle_15",
"situation": "CROSS_WAIT",
"action": "SLIGHTLY_LEFT_SLOWER"
}
```

