# OpenReview forum: "Large Language Models as Decision Makers for Autonomous Driving"
_ICLR.cc/2024/Conference — Submitted to ICLR 2024_

### Official Review · Reviewer_hRfh · 2023-10-17

**Soundness:** 3 good
**Presentation:** 3 good
**Contribution:** 3 good
**Rating:** 6
**Confidence:** 3

**Summary:**

This paper leverages LLM as a decision maker for controlling vehicles in complex AD scenarios. In particular, it proposes a chain-of-thought framework for LLMs for driving scenarios. and techniques (attention allocation, situational awareness, multi-vehicle joint control) for directing low-level controllers. To evaluate their methods, the authors compare it against MPC and RL baselines in five scenarios, and show superiority of the proposed methods. Besides, the proposed method has been shown to be able to regulate driving behavior via text input.

**Strengths:**

1.important and timely topic

The paper is on one of the most popular topics in the field of autonomous driving: LLM for driving.

2.practical framework/methods with moderately high potential impact

The proposed framework can be beneficial for future work in the field of LLM for driving. In particular, the proposed attention allocation and situational awareness are simple but practical approaches for guiding LLM via injecting a kind of structure based on human knowledge.

3.well-written

The diagrams and examples clearly demonstrate the methodology and the superiority of the proposed method over the baseline methods.

**Weaknesses:**

1.did not compare with strong baselines for planning

The baselines are very limited and no strong baselines on trajectory planning (e.g., Wayformer [1]) or traffic simulation (e.g., BITS [2]) have been compared with. This potentially makes one doubt how useful the proposed method can be in practice.

[1] Wayformer: Motion Forecasting via Simple & Efficient Attention Networks, Nigamaa Nayakanti, Rami Al-Rfou, Aurick Zhou, Kratarth Goel, Khaled S. Refaat, Benjamin Sapp

[2] BITS: Bi-level Imitation for Traffic Simulation, Danfei Xu, Yuxiao Chen, Boris Ivanovic, Marco Pavone

2.the use of manually designed situations can potentially limit the scalability of the proposed method

It seems that the proposed method has a fixed number of situations manually designed which potentially limit the generalization and scalability of the proposed method.


3.Missing existing/concurrent relevant work on LLM for traffic generation/self-driving

I suggest the authors discuss the following missing literatures in the related work.

(1) literatures on language controlled traffic generation:

Language-Guided Traffic Simulation via Scene-Level Diffusion, Ziyuan Zhong, Davis Rempe, Yuxiao Chen, Boris Ivanovic, Yulong Cao, Danfei Xu, Marco Pavone, Baishakhi Ray

(2) literatures on LLM controlled self-driving:

DriveGPT4: Interpretable End-to-end Autonomous Driving via Large Language Model, Zhenhua Xu, Yujia Zhang, Enze Xie, Zhen Zhao, Yong Guo, Kwan-Yee. K. Wong, Zhenguo Li, Hengshuang Zhao

GPT-Driver: Learning to Drive with GPT, Jiageng Mao, Yuxi Qian, Hang Zhao, Yue Wang

DiLu: A Knowledge-Driven Approach to Autonomous Driving with Large Language Models, Licheng Wen, Daocheng Fu, Xin Li, Xinyu Cai, Tao Ma, Pinlong Cai, Min Dou, Botian Shi, Liang He, Yu Qiao

4.limitations are not discussed

**Questions:**

-How many X^t is designed? What are all the situations you consider?

-How A^t influences bias?

-Can you provide the details for the metrics used in Table 2?

---

### Official Review · Reviewer_aTEF · 2023-10-28

**Soundness:** 2 fair
**Presentation:** 2 fair
**Contribution:** 1 poor
**Rating:** 3
**Confidence:** 4

**Summary:**

In this paper, the authors present the idea of using Large Language Models (LLMs) as decision-makers in autonomous driving scenarios. The motivation behind this idea is to utilize the common sense emerging in LLMs like GPT-3.5 to guide autonomous vehicles. The use of LLM, the authors posit, would make decision-making more interpretable as reasoning is done in human-understandable natural language. The architecture of the overall system is as follows: A prompt describing the road scene is given to the LLM, LLM outputs a high-level decision, this decision is converted to a set of matrices using pre-defined rules, the matrices are then passed to a model predictive control (MPC) which provides an executable action. The authors conduct their experiments on GPT-3.5 in an IdSim road simulation environment. Multiple driving scenarios are chosen for study, and the LLM + MPC system is compared against RL and MPC baselines.

**Strengths:**

I find the following strengths in the paper:

The motivation for using LLM for decision-making in autonomous driving is clearly a promising direction, given the model's language-based decision-making. The justification for better interpretability follows, too.

**Weaknesses:**

Fundamental issues:
1. Sufficiency of language-based autonomous driving system: I find it difficult to accept that only language information is enough to generate efficient suggestions for driving. Driving usually involves visual and spatial cues, too, and they are too important to be left out of the discussion.
2. Toy nature of IdSim environment: The setup used to test the ideas is quite preliminary to derive any conclusions. The paper uses around 25 driving scenarios, although it does not mention how these scenarios are selected. Also, the environment being too simplistic, it is very hard to make generalizations using the results observed in this setting. The language-only input makes sense only in this toy setup; real-world autonomous driving would require a multi-modal language model.
3. Baselines are too weak: The paper mentions that their approach is evaluated against RL and MPC baselines. However, no details are provided about their implementations. Also, it is unclear on what kind of input the algorithms are run, as in, there is no clarity on whether the RL algorithm had the same information as the LLM approach.
4. Unavailability of code: The evaluation of the overall experimental setup becomes even harder, given the lack of code implemented. Also, it seems difficult for me to get the same results as shown by the paper due to a lack of clarity on how the MPC matrices are designed.
5. Logical reasoning capability of LLMs: It has been pointed out multiple times that LLMs have limited logical reasoning abilities. The current paper claims to use the emergent reasoning of LLMs for driving. The scenarios presented in the paper for testing LLMs' abilities do not seem to require any reasoning as such, or I could be missing something here.
5. Writing lacks clarity:
	- Figures are not well-labeled and do not provide a clear description of the flow. For instance, in Figure 1, it is unclear whether the 'common sense' is fed to the LLM or considered an emergent phenomenon. In Figure 2, it is unclear what 'judge', 'select', and 'select' exactly mean.
	- Notations are hard to follow. If $S^t_i$ tracks individual vehicles, what does $env^t$ represent? Why is it dependent on time? From the description, it occurs to me that the environment in the road scene is static; only vehicles move. Even the video demonstrations on the website confirm this.

**Questions:**

In lieu of the above-mentioned weaknesses, I have the following questions for the authors:
1. Why is language sufficient to drive an autonomous car? In case we wish to bring in the visual cues, how can we incorporate them into the proposed architecture?
2. Which RL algorithm is implemented as the baseline? What are the agent's inputs? Does it see the same information as provided to LLM for generating high-level decisions?
3. Would it be possible for the authors to share the code used to perform the experiments?
4. What are the different logical reasoning capabilities tested during the experiments? The paper says it mainly focuses on turning left. Doesn't it limit the generalization of LLM's abilities to other cases?

---

### Official Review · Reviewer_C5TQ · 2023-10-29

**Soundness:** 2 fair
**Presentation:** 3 good
**Contribution:** 2 fair
**Rating:** 3
**Confidence:** 4

**Summary:**

Current autonomous driving algorithms still face some challenges such as comprehending high-level information, generalizing to rare events, and providing interpretability. Large language models (LLMs) show great potential in solving problems with human common sense and providing human-understandable explanations. Therefore, this paper proposes to use LLM as a high-level planner and use MPC as a low-level controller. The authors evaluate their driving pipeline in both single-vehicle tasks and multi-vehicle coordination. Results show that their method consistently surpasses baselines.

**Strengths:**

1. Combining LLM and MPC is a smart way to leverage the high-level reasoning capability of LLM. Since LLM is not good at low-level control as shown in previous literature, using MPC to output low-level actions helps incorporate strong prior knowledge into the whole pipeline.

2. Rare and long-tail events put great challenges to data-driven algorithms. Using common sense to solve such long-tail problems is a promising way. Since LLM already contains a lot of common sense, this paper is a good attempt to explore this direction.

3. The paper is well-written and easy to follow. Figure 2 provides a clear illustration of the entire pipeline, which makes me easy to understand the method proposed in this paper.

**Weaknesses:**

1. A common question for LLM-based agents is the trustworthiness. When used for safety-critical applications like autonomous driving, the requirements for safety are much higher than QA or code generation tasks. As far as I know, GPT3.5 and GPT4 have uncontrollable randomness even with temperature = 0. Does this randomness influence the running results? How to ensure that LLM always outputs reasonable results. Are the results consistent if the version of the ChatGPT model changes?

2. How about the running frequency of the entire system? Since driving requires a high frequency to make decisions, the speed of processing the information with language may not be fast enough. This is one of the biggest challenges of using LLM for driving, which may be unaffordable and unsolvable with a general-purpose LLM. Using LLM which can only communicate with downstream modules with language wastes a lot of time on extracting information.

3. In the first paragraph of the introduction, the authors provide an example “Human drivers intuitively know that according to traffic rules, they should slow down and yield, even if it is technically possible to speed through”. This example is confusing to me as it just shows two types of common driving styles of human drivers (aggressive and conservative styles). I am not aware of any complex rules or reward functions here since it is easy to force the driving algorithm to be conservative with some simple rules. Actually, considering the complexity of the pipeline designed in Figure 2, I am not sure which one is more complex. Maybe several simple cost functions in an MPC can already solve the scenarios in the experiment part.

4. It seems that one important motivation for using LLM for driving is solving rare and long-tail events. However, I only find Figure 1 discusses one specific case where a breakdown vehicle is stopped in the middle of the road in a roundabout. I can’t find any evidence or experiment results showing that LLM broadly solves long-tail problems.

5. In the evaluation part, only one RL algorithm (not mentioned which algorithm exactly) and one MPC are compared. Since modern AV algorithms are usually complex, I don’t think such a comparison is not convincing enough to demonstrate the advantage of the LLM method.

**Questions:**

1.	Could the authors report the running time of all methods in the experiments?
2.	There are 5 scenarios mentioned in Table 1 but no detailed explanation of them. I cannot evaluate the realism and difficulty of them.
3.	Could the authors explain more about the interpretability of the LLM method? Does this interpretability mean the reason for decision-making? Why is such interpretability useful and can we always trust such interpretability? If the interpretability is something shown in Figure 1, I think all collision avoidance algorithms can provide the reason for the decision.

---

### Official Review · Reviewer_aAFA · 2023-10-29

**Soundness:** 2 fair
**Presentation:** 2 fair
**Contribution:** 2 fair
**Rating:** 3
**Confidence:** 3

**Summary:**

This paper focuses on using LLMs for autonomous driving, in which a sequence of prompts allows an LLM to provide driving actions to an ego-vehicle in specific driving scenarios.

**Strengths:**

- interesting application to showcase the power of LLMs.
- demonstration of a full pipeline on how to use an LLM for control from observations.

**Weaknesses:**

-the paper lacks a lot of important details. This work is definitely not reproducible as is. Information about how to go from LLM output to MPC, details about the different situations with predefined weight matrices. Details about these values and parameters are missing and are not present even in the supplementary material.
- looking at the prompts and the LLMs answers in the appendix, it seems that there are a lot of suggestions of what to do and what each action will do (cross normal/wait/fast)

**Questions:**

- I would like more information about scenarios. It is not clear to me if the other agents are reactive to the ego vehicle or purely replaying their trajectories. how long are these scenarios? what are the possible actions? it seems there are only predefined discrete actions to choose from?
- What does it mean that the LLM chooses a vehicle? (see appendix B) Isn't the LLM controlling the ego vehicle? It is unclear if the LLM is controlling ego or vehicle_15.
- what are the "predefined situations" you mention that the MPC relies on?

---

### Meta-Review · Area_Chair_Kfky · 2023-12-05

**Metareview:**

This work seeks to use LLMs as the core decision making component in an autonomous driving setup, which is an interesting direction given LLMs are able to grasp higher level concepts. The idea is novel however the reviewers had many concerns about both the method itself and the capabilities of the resulting method with respect to inference speed, safety and consistency of results. These concerns likely could have been addressed in the rebuttal phase but the authors did not respond, which is unfortunate.

**Justification For Why Not Higher Score:**

The reviewers had a long list of clarity concerns which seemed fixable but the authors made no attempt to address them.

**Justification For Why Not Lower Score:**

N/A

---

### Decision · Program_Chairs · 2024-01-16

Reject